# SIZE study: study protocol of a multicentre, randomised controlled trial to compare the effectiveness of an interarcuair decompression versus extended decompression in patients with intermittent neurogenic claudication caused by lumbar spinal stenosis

Jamie Arjun Sharma ,[1] Pravesh S. Gadjradj ,[2] Wilco C. Peul,[2] Maurits W. van Tulder ,[3] Wouter A. Moojen,[2,4] Biswadjiet S. Harhangi,[1] on behalf of the SIZE-study group

For numbered affiliations see end of article.

**Correspondence to**
Dr Biswadjiet S. Harhangi; b.s.harhangi@erasmusmc.nl

## ABSTRACT

**Introduction** Intermittent neurogenic claudication (INC) is often caused by lumbar spinal stenosis (LSS). Laminectomy is considered a frequently used surgical technique for LSS. Previous studies have shown that laminectomy can potentially cause lumbar instability. Less invasive techniques, preserving midline structures including the bilateral small size interarcuair decompression, are currently applied. Due to lack of evidence and consensus, surgeons have to rely on their training and own experiences to choose the best surgical techniques for their patients. Hence, an observer and patient blinded multicentre, randomised controlled trial was designed to determine the effectiveness and cost-effectiveness of bilateral interarcuair decompression versus laminectomy for LSS.

**Methods and analysis** 174 patients above 40 years with at least 12 weeks of INC will be recruited. Patients are eligible for inclusion if they have a clinical indication for surgery for INC with an MRI showing signs of LSS. Patients will be randomised to laminectomy or bilateral interarcuair decompression. The primary outcome is functional status measured with the Roland-Morris Disability Questionnaire at 12 months. Secondary outcomes consist of pain intensity, self-perceived recovery, functional status measured with the Oswestry Disability Index and a physical examination. Outcome measurement moments will be scheduled at 3 and 6 weeks, and at 3, 6, 12, 18, 24, 36 and 48 months after surgery. Physical examination will be performed at 6 weeks, and 12, 24 and 48 months. An economic evaluation will be performed and questionnaires will be used to collect cost data.

**Ethics and dissemination** The Medical Ethical Committee of the Erasmus Medical Centre Rotterdam approved this study (NL.65826.078.18). The results will be published in an international peer-reviewed journal.

## Strengths and limitations of this study

► Patient and observer blinded randomised controlled design.
► Both patient-reported outcomes and objective functional test outcomes will be measured.
► Inclusion of an economic evaluation.
► Instability as outcome may not be measurable in 2 years of follow-up.
► Diversity in terminology on posterior decompression techniques may lead to variability on both surgical techniques.

**Trial registration number** ClinicalTrials.gov (NCT03480893).
**IRB approval status** MEC-2018-093.

## INTRODUCTION

Lumbar spinal stenosis (LSS) is described as the narrowing of the diameter of the spinal and nerve root canals, which can cause compression of the neural structures in the canals.[1] LSS is a common medical condition in the ageing population and is considered the most common reason for lumbar spine surgery for people over the age of 65.[2–4] Intermittent neurogenic claudication (INC) is the most important symptom of LSS, with leg pain and heaviness of the legs, usually bilateral, which exacerbate during standing and walking. Apart from leg pain, back pain is often also present.[5] As the symptoms aggravate, the ability to walk can become considerably limited, which is the



dominant functional impairment and the most common reason for seeking medical care.[6 7]

Conservative treatment, such as physical therapy and pain medication, is the first treatment option and may give some relief of symptoms.[8] Previous research shows that there is sparse evidence of the effectiveness of surgery compared with non-operative treatment.[9] However, usually when conservative treatment fails, surgical treatment is considered the next option.[10 11] The first technique described to widen the lumbar spinal canal is the laminectomy technique, also known as wide bony decompression, which is a widely used surgical technique for patients with INC caused by LSS.[12] Given that patients often experience postoperative back pain after laminectomy, it is hypothesised that laminectomy is a ground for potential lumbar instability and iatrogenic scoliosis.[13–15] Performing a procedure with potentially more complications in an elderly population could be considered as doubtful. Hence, less invasive techniques, such as the bilateral small size interarcuair decompression, otherwise known as a bilateral laminotomy, limited bony decompression or interlaminar decompression, were developed and implemented in clinical practice. These techniques require less bone tissue to be removed, which possibly can reduce the prevalence of lumbar instability in these patients and consequently improve functioning.[16] Furthermore, it is believed that, compared with laminectomy, these less invasive techniques have the ability to decompress the nerves while preserving the spinal integrity. However, the risk of an insufficient decompression may be higher, possibly resulting in reoperations due to residual stenosis.[16–18]

The presumption is often made that, after a wide decompression, recurrence of the complaints is scarce. The opinions on this matter are diverse and subject to debate.[19] Therefore, a randomised controlled trial (RCT) is needed to clarify whether bilateral interarcuair decompression is effective and cost-effective compared with a classical laminectomy.

## Study objectives
The primary objective of this study is to determine whether bilateral interarcuair decompression is more effective than laminectomy in patients with INC caused by LSS. The secondary objective is to evaluate if bilateral interarcuair decompression is more cost-effective than laminectomy.

## Study design
The study is designed as a multicentre, patient and observer blinded RCT with an economic evaluation alongside. Participants will be allocated to one of the two groups: laminectomy or interarcuair decompression.

## METHODS
### Study setting/population
In total 174 patients with an MRI-confirmed LSS will be recruited. All patients who will be included should have an indication for surgery due to LSS with minimally 12 weeks of INC, as is customary in clinical practice, and failed conservative treatment. Patients will be recruited from two hospitals and one private health clinic in the Netherlands:

► Erasmus Medical Centre (MC) in Rotterdam.
► Maasstad Hospital in Rotterdam.
► Park MC in Rotterdam.

## Eligibility criteria
In order to be suitable to participate in this study, patients must meet all of the following criteria:

### Inclusion criteria
► At least 12 weeks of INC with a leg pain level above 3 on a Numeric Rating Scale (NRS), without sufficient response to conservative management.
► Indication for an operation according to consensus of surgeon.
► MRI demonstrating LSS.
► Age above 40 years.
► Sufficient knowledge of the Dutch language in order to comprehend the questionnaires and patient information.
► Written informed consent.

### Exclusion criteria
Patients who meet any of the following criteria will be excluded from participating in this study.
► History of lumbar spine surgery.
► More than two lumbar levels needing surgery.
► Degenerative spondylolisthesis greater than Meyerding grade I (on a scale of I–V).[20 21]
► Concomitant scoliosis or disc herniation.
► Severe comorbid medical disorder (American Society of Anesthesiologists >3).
► Serious psychopathological disorder.
► Pregnancy.
► Active malignancy.
► Plans to move abroad during study period.

Eligible patients will be randomised in a 1:1 ratio to the two groups.

## Interventions
### Intervention: bilateral small size interarcuair decompression
The intervention consists of the interarcuair decompression, otherwise known as a bilateral laminotomy, limited bony decompression or an interlaminar decompression. General anaesthesia is to be administered. A midline skin incision is made, after which the paravertebral muscles are dissected subperiosteally and retracted bilaterally. Decompression will be applied through decompression of the ligamentum flavum and partial laminotomy if necessary. The lateral recess will be opened bilaterally and a partial medial facetectomy will be performed to decompress neuronal structures including the exciting nerve root. The lamina will be removed up to the ligamentum flavum. Interspinous ligaments are spared.[22] The wound is closed in layers. Patients are operated with a loupe magnification or microscope depending on the surgeon's preference.

## Control group: conventional laminectomy

The control group consists of conventional laminectomy, otherwise known as wide bony decompression. General anaesthesia is to be administered. A midline skin incision is made over the spinous processes. The laminae of the affected level(s) are exposed subperiosteally, and the supraspinous ligament is incised. The spinous process is removed. The supraspinous and interspinous ligament of the affected level is removed by drill or Kerrison punched. Both laminae are removed at the affected level, leaving the facet joint intact. The lateral recess is opened bilaterally and a partial medial facetectomy will be performed to decompress neuronal structures including the exciting nerve root.[16] To clarify, when a single level stenosis is present (eg, L4–L5), both laminae L4 and L5 will be removed. When a double level stenosis (eg, L3–L4 and L4–L5) is present, three laminae (L3, L4 and L5) will be removed. The wound is closed in layers with or without a suction drain. Patients will be operated with loupe magnification or microscope depending on the surgeon's preference.

## Use of co-intervention

Pain medication will be provided to patients after surgery, should this be necessary. Further, the use of co-interventions will be tracked by cost questionnaires, in which medication usage and any healthcare utilisation is monitored throughout the follow-up period.

## Outcomes

### Primary outcomes

The primary outcome is a functional status measured using the Roland-Morris Disability Questionnaire (RMDQ). This is a 24-point questionnaire that is designed to assess the functional status in patients with low back pain. This outcome measure has been identified as one of the most commonly used outcomes in a population with chronic low back pain.[23–27]

### Secondary outcomes

#### Oswestry Disability Index

The Oswestry Disability Index (ODI) is one of the principal condition-specific outcome measures used in the management of spinal disorders.[28] The current V.2.1a will be used.[29] It consists of 10 questions, each with 6 possible answers and each answer option receives a score of 0–5 points, yielding a score range between 0 and 50, which is scaled to a 100% range. The questions focus on a range of daily physical functions and how the back or leg pain is affecting the patient's ability to manage in everyday life.

#### NRS for leg and low back pain

This parameter will measure the experienced pain intensity in both legs (affected and non-affected) and back. Pain intensity will be rated on an 11-point scale, varying from 0 representing 'no pain' to 10 'the most terrible pain imaginable'. Patients are not permitted to see the pain score indicated during previous visits. Reliability and validity of NRS have been shown.[30]

### Short form 36

The quality of life will be measured with the Dutch version of the Short Form 36 (SF-36). The SF-36 has been applied and successfully validated in populations with low back pain.[31] This questionnaire relates to the analysis of the general functional status of the patient. The questionnaire is divided into eight health concepts: (1) physical functioning, (2) role limitations because of physical health problems, (3) bodily pain, (4) social functioning, (5) general mental health (psychological distress and well-being); (6) emotional role limitations; (7) vitality and (8) general health perceptions. The scores of the different health concepts are added up into a scale of 0–100. A higher score reflects a better general health condition. The physical and mental component summary will be assessed separately.

### Patients' perceived recovery

Recovery will be measured using a 7-point Likert scale. The score on this scale varies from 'complete recovery' to 'worse than ever'. The outcomes 'complete recovery and 'almost complete recovery' will be dichotomised and considered as recovered.[32]

### Patients' satisfaction

Patients' satisfaction of change and satisfaction of treatment will be assessed using a 7-point Likert scale, ranging from 'completely satisfied' to 'completely dissatisfied'. 'Completely satisfied' and 'almost completely satisfied' will be defined as good outcome.

### Physical examination

A physical examination will be performed before surgery, at 6 weeks, 12, 24 and 48 months after surgery. This examination will include a brief neurological examination, the Timed Up and Go Test (TUG), the Timed Chair-Stand Test (TCST), the 6-Minute Walking Test (6MWT) and patient's weight.

A. Neurological examination

The neurological examination will include the muscle strength of the quadriceps, the iliopsoas, the hamstrings, the gastrocnemicus and the tibialis anterior muscle, which will be noted as a difference in strength between both legs. We will evaluate if there is a difference in muscle strength between the affected leg and the other leg.

B. TUG

Patients are asked to sit in a chair with their arms resting on the armrests. On direction of the research nurse, patients have to walk as fast as possible (without running) to a wall at 3 m distance of the chair. When they have reached that wall, a turn of 180° is made and they walk directly back to the chair and sit down. The research nurse will measure the time (in seconds) it takes for a patient, to perform this action. The time is stopped when the patients are seated again.[33] Patients need to perform this test, three times. Patients are allowed to wear their regular shoes and



use a walking aid, if necessary. The TUG test has been described as a valid, reliable tool in the assessment of objective functional impairment in patients with lumbar degenerative disc disease.[34 35] However, this has not been done for LSS. The validation for LSS will be conducted in this study.

C. TCST

Patients are asked to sit down on an armless chair, firmly placed against a wall, with their arms folded across their chest. They are instructed to place their feet flat on the ground. To become familiarised with the exercise, the patients are asked to stand up and sit down again without using their arms. If assistance is required during this action, or this movement cannot be completed by the patient, the test is abandoned. If the manoeuvre is possible, the patient will be asked to, on the word 'go', stand up entirely and sit down again, five times as quick as possible. The research nurse will time the five movements in seconds from the command 'go' to the completed fifth stand. The measured time will be noted as the score. If the patient is unable to perform the test five times, a maximum score will be noted of 30 s.[36 37]

D. 6MWT

This test is performed to assess the gait pattern, walking speed and physical endurance of patients. The patient will be requested to walk at such a speed that at the end of 6 min, the patient will have the feeling of maximum output.[38] Before taking the test, the patient will be given the instructions to install the 6MWT application on their cell phone.[39 40] During these 6 min this application will measure the walking distance. During the walking test, the patient is permitted to use a walking aid and/or orthosis which will be noted.

### Functional lumbar X-rays

Functional X-rays will be obtained from all patients. The X-ray will be made standing from Antero-posterior and lateral position to assess spondylolisthesis. X-rays will also be made in maximum flexion and extension position. Degenerative spondylolisthesis is defined as a vertebral slip of at least 3 mm.[41] After the measurement of the vertebral slip, this will be classified according to the Meyerding classification. This will be obtained at 6 weeks postoperatively, for level verification and for verification of correct procedure and after 2 years postoperatively to look for possible instability.

### EuroQol-5-dimension-5-levels

The EuroQol-5-dimension-5-levels (EQ-5D-5L) is a validated instrument to assess health-related quality of life. The tool measures five dimensions:
► Mobility.
► Self-care.
► Pain/discomfort.
► Daily activities.
► Anxiety.

Each dimension consists of one item, in which five levels are distinguished (no problems, minor problems, moderate problems, major problems, severe problems, unable to). The Dutch tariff of the EQ-5D-5L will be used to calculate the quality-adjusted life years (QALYs).[42 43]

### Cost-diaries

Based on a diary with made costs, which will be completed by patients, insight will be obtained in the need for care, working participation and direct and indirect medical cost. Direct medical cost will be estimated on basis of the cost centre method. In addition it is expected that the study population will consist of elderly people who may be retired. For the cost calculation additional costs, such as help in house, transport and help from family will be taken into account.

For an estimation of direct and indirect cost, permission of the patient will be asked to request the total amount of cost incurred during the research period from the insurance company. No consideration by participation in the investigation as compared with daily practice will be deducted. The additional investigational costs include mainly the additional visits to the outpatient clinic.

The patient will be requested to keep a diary for the financial aspects of the consequences of INC and corresponding treatment. The patient will be requested to record the following items:
► Visits to general practitioner.
► Visits to physiotherapist.
► Visits to specialist.
► Alternative medicines and devices (eg, rollator).
► Number of days of hospitalisation.
► Pain medication; dosage and frequency.
► Illness-related days of absence at work, if any.
► Cost of loss production and substitute manpower, if any.
► Additional travelling expenses on account of INC.
► Help in house holding.

### Complication and reoperation incidence

Immediately after surgery, the surgeon will report any perioperative complications such as cerebrospinal fluid leakage, nerve root damage and if the operation was implemented at the wrong vertebral level. Additionally, a systematic assessment of complications (including urinary tract infection, progressive neurological deficit and secondary bleeding) will be carried out by the surgeon and the research nurse shortly after discharge.

Reoperation is considered a bad outcome. The incidence of reoperation as well as perioperative morbidity will be assessed in both groups, using surgical parameters such as blood loss, time of surgery and length of hospital stay.

We used the Standard Protocol Items: Recommendations for Interventional Trials checklist when writing our report.[44]

## Patient and public involvement

No patients involved.

## Sample size

The sample size of this study is calculated for the primary endpoint of group difference in the RMDQ at 12 months. According to literature, results obtained after interarcuair decompression should be 3 points lower on the RMDQ compared with those for the laminectomy group in order to detect superiority.[45]

Hence, the sample size of this study is based on a superiority design, using a delta of 3, an assumed true mean difference of 0 and a pooled SD of 6. Using an alpha of 0.05, and a power of 90%, we calculated a sample size of 69 patients per group. Accounting for a 20% loss in follow-up, we will therefore enrol 174 patients (87 patients per group).

## Allocation

Computerised block randomisation will take place after the patient has been anaesthetised. Allocation concealment will be assured by using ALEA, a web-based data system supervised by the Clinical Trial Centre of the Erasmus MC. Variable block sizes of 4, 6 and 8 will be used and stratified by treatment centre. Patients will be randomised in a 1:1 ratio to laminectomy or bilateral interarcuair decompression. Each patient will be given a unique study number.

## Blinding

All researchers and patients will be blinded for the treatment. This is possible, because there are no fundamental differences between both procedures (eg, general anaesthesia, spinal incision, inpatient procedure). The statistical analyses will be performed blinded. The surgeon cannot be blinded.

## Data collection methods

All patient reported outcomes will be collected preoperatively and after 3 weeks, 6 weeks, 3 months, 6 months, 12 months, 18 months, 24 months, 36 months and 48 months (see table 1). Patients will receive an email with a link, reminding them of the upcoming data collection and requesting them to fill in the web-based questionnaires. If the questionnaires are not completed, the patient will receive a reminder after 1 and 2 weeks. After 2 weeks, non-responders will be contacted by phone.

## Data management

Data from initial visits to the hospital, follow-up visits and questionnaires will be entered into a database via an electronic data system (Gemstracker). This data will be noted and analysed by using coded information (study numbers) without any personal identifiers. Data will be stored via the secure data management system of the trial coordination centre of the Erasmus MC.

## Statistical analysis

Patient characteristics will be summarised per treatment group to determine whether the randomisation was balanced. Continuous variables will be presented using means and SDs, or in the case of non-normally distributed variables using medians and IQRs. Normality will be graphically assessed. Categorical variables will be

| Table 1 | Participant timeline schedule | | | | | | | | | | | |
|---|---|---|---|---|---|---|---|---|---|---|---|---|
| | **Visit plan and case report forms** | | | | | | | | | | | |
| **Table** | **Intake** | **Surgery (t$_0$)** | **1–2 day(s) following surgery** | **FU 3 weeks** | **FU 6 weeks** | **FU 3 months** | **FU 6 months** | **FU 12 months** | **FU 18 months** | **FU 24 months** | **FU 36 months** | **FU 48 months** |
| Visit surgeon | x | | | | x | | | | | | | |
| Research nurse visit | x | | | | x | | | x | | x | | x |
| Randomisation | | x | | | | | | | | | | |
| Surgery | | x | | | | | | | | | | |
| Discharge | | | x | | | | | | | | | |
| Physical examination | x | | | | x | | | x | | x | | x |
| Functional X-rays | x | | | | x | | | | | x | | |
| Roland-Morris Questionnaire | x | | x | x | x | x | x | x | | x | x | x |
| Oswestry Disability Index | x | | | x | x | x | x | x | | x | x | x |
| NRS leg and back pain | x | | x | x | x | x | x | x | | x | x | x |
| Patient self-received recovery and satisfaction | x | | x | x | x | x | x | x | | x | x | x |
| EuroQol-5D | x | | x | x | x | x | x | | | x | x | x |
| Cost questionnaires | x | | x | x | x | x | x | x | x | x | | |
| Short form 36 | x | | x | x | x | x | x | | | x | x | x |
| Revisit and complications | With occurrence | | | | | | | | | | | |

FU, follow-up; NRS, Numeric Rating Scale.



presented using counts and percentages. The primary analysis will study differences in the primary outcome measure (RMDQ). The primary endpoint of interest is 12 months after randomisation. The results of the RMDQ will be assessed using generalised linear mixed models (GLMM), to account for the repeated measurements within patients and the multicentre stratification.

The secondary outcomes (ODI, NRS leg and back pain score, TUG test, TCST and the SF-36) will be similarly assessed using GLMMs and will be analysed for exploratory purposes. In all analyses the primary assessment of treatment effect will be the estimate of the main effect within the appropriate model at 12 months, adjusted for the stratification factors and main covariates. Time to recovery will be analysed using survival analysis (Cox proportional hazards). At 24, 36 and 48 months, the GLMM repeated-measurement analyses using the compound symmetry covariance structure will be used while *group* (interarcuair and conventional), *time* (moments of measurement) and the *interaction* between group and time (grou×time) will be entered as independent variables. Dependent variables are the primary and secondary outcome parameters. Likert scales for self-reported patient recovery and satisfaction will be assessed using descriptive statistics (counts and percentages) and appropriate plots. Comparisons of Likert scale results at individual time points will be tested using Mann-Whitney U tests. Complications and incidence of reoperations will be assessed using descriptive statistics (counts and percentages). Comparisons between treatment groups for specific complications of interest will be assessed using $X^2$ tests or Fisher's exact tests. If necessary, missing data will be imputed using multiple imputation techniques.

## Economic evaluation
An economic evaluation will be performed from the societal and healthcare perspectives to evaluate the cost-effectiveness of the bilateral interarcuair decompression compared with a 'classic' laminectomy for patients with INC. When the societal perspective is applied, all costs and consequences relevant to the interventions will be taken into account, whereas only costs accruing to the formal Dutch healthcare sector will be considered when the healthcare perspective is applied. Intervention costs will be estimated using a micro-costing approach. Cost questionnaires will be administered at 3, 6 and 12 weeks covering the entire period and at 6, 12, 18, 24, 36 and 48 months after surgery covering the previous 3 months. Data will be collected on healthcare utilisation, the use of informal care, absenteeism, presenteeism and unpaid productivity losses. Costs will be valued using guideline prices recommended in the Netherlands.[46]

A cost-utility analyses will be conducted with QALYs as outcome. For estimating QALYs, the patients' EQ-5D-5L health states will be converted into utility scores using the Dutch tariff.[43] Linear interpolation between measurement points will be used to subsequently calculate QALYs. Missing data will be imputed using multiple imputation

by chained equations.[47] An incremental cost-utility ratio will be calculated by dividing the difference in costs by the difference in effects. The cost-utility ratio expresses the incremental costs per QALY gained. Bootstrapping techniques will be used to estimate the uncertainty surrounding the cost-effectiveness estimates. Uncertainty will be shown in cost-effectiveness planes and cost-effectiveness acceptability curves, and sensitivity analyses will be performed to test the robustness of the study results.

## Data monitoring
This study will be monitored according to a detailed monitoring plan adapted to the risk classification of the Dutch University Federation guidelines. Based on this guideline, the risk classification of this study is regarded negligible. Considerations in this assessment are that this is an investigator-initiated trial, not with vulnerable patients, and while side effects, such as nerve root damage, are known, severe adverse events (AEs) are extremely rare. Audits may be required by the Medical Ethical Committee or by the regulatory authority inspections and will be granted if necessary. Patients' permission for these audits is obtained with informed consent.

## ETHICS AND DISSEMINATION
### Research ethics approval
This protocol, along with the informed consent forms, recruitment materials and other requested documents, was reviewed and approved by the Medical Ethical Committee of Erasmus MC in Rotterdam with respect to scientific content and compliance with applicable research and human subject regulations.

### Protocol amendments
Any subsequent amendments will be reviewed and approved by the ethical review bodies.

### Consent or assent
Surgeons will introduce the trial to patients at the outpatient clinic and will hand them a patient information brochure regarding the main aspects of the trial. Patients will be given 5 working days to decide if they want to participate in the trial. Trained research nurses will call patients after 1 week to ask if they wish to partake in the trial. After verbal consent, an appointment will be made with the research nurse, where informed consent forms will be obtained (see online supplemental appendix 1).

### Confidentiality
All study-related information will be stored securely at the study site. All participant information will be stored in locked file cabinets with limited access. All reports, data collection (eg, case report forms) and administrative forms will be entered into an online data system. All these documents will be identified by a coded study number only to maintain the patient confidentiality. These data will be stored for at least 15 years.

## Ancillary and post-trial care

AEs and serious AEs will be monitored. SAE will be reported within 24 hours. The sponsor has an insurance which is in accordance with the legal obligations in the Netherlands. The insurance applies to the damage that becomes apparent during the study or within 4 years after the end of the study.

## Dissemination policy

The final trial results will be communicated to the participants, healthcare professionals, professional organisations and relevant guideline committees in the Netherlands. The results will be published in an international peer-reviewed open-access scientific journal. There are no publication restrictions.

## DISCUSSION

In this article, a protocol for a multicentre patient and observer blinded RCT of the SIZE study is presented. Surgical treatment, such as laminectomy or interarcuair decompression, for LSS has been shown to reduce symptoms.[10 11] Due to an abundance of surgical interventions to treat LSS, the surgical management as of now, demonstrates a wide variety of preferred treatments by spine surgeons.[48] To reduce this variety, there is a necessity for RCTs, to create guidelines on the optimal treatment for LSS.[49] There is an ongoing trial that compares three surgical decompression techniques in patients with lumbar spinal stenosis. Yet, conventional laminectomy is not applied in this trial, which is still a frequently used surgical technique.[48 50] A recent study compared the radiological and clinical results of bilateral interlaminar decompression and laminectomy.[51] However, this study was a single-centre study, was not blinded, and did not include a computerised randomisation. Furthermore, there were no data regarding the cost-effectiveness of both surgical interventions. The direct and indirect costs of an intervention are important determinants, that have to be taken into account by the surgeon, during everyday decision-making. In the SIZE study, we intend to include an economic evaluation to estimate the cost-effectiveness and therefore acquire new information on this topic.

**Author affiliations**
[1]Neurosurgery, Erasmus Medical Center, Rotterdam, The Netherlands
[2]Neurosurgery, University Neurosurgical Center Holland, Leiden University Medical Center and The Hague Medical Center, Leiden, The Netherlands, Leiden, The Netherlands
[3]Health Sciences, University of Amsterdam, Amsterdam, The Netherlands
[4]Neurosurgery, Medical Centre Haaglanden, Den Haag, The Netherlands

**Acknowledgements** The SIZE-study group consists, except for the authors, of Dr H.P. van Putten and Dr R. van Rolde. The authors would furthermore gratefully acknowledge Katya Mauff for statistical advice.

**Collaborators** The SIZE-study group: Katya Mauff, Hendrikus van Putten.

**Contributors** JAS and PG contributed to drafting the article and took the lead in writing the manuscript. All authors provided critical feedback and helped shape the research and manuscript. JAS, PG, WAM, WP and BH devised the main conceptual ideas and designed the study. MWvT designed the statistical analysis and the economic evaluation. PG and BH provided funding for conducting this study. JAS, PG, WAM, WP, MWvT and BH contributed to the study protocol.

**Funding** This work was supported by MRace grant, number 2017-17106 (Institutional Grant of Erasmus Medical Centre).

**Competing interests** None declared.

**Patient and public involvement** Patients and/or the public were not involved in the design, or conduct, or reporting, or dissemination plans of this research.

**Patient consent for publication** Not required.

**Provenance and peer review** Not commissioned; externally peer reviewed.

**ORCID iDs**
Jamie Arjun Sharma http://orcid.org/0000-0003-2588-2153
Pravesh S. Gadjradj http://orcid.org/0000-0001-9672-4238
Maurits W. van Tulder http://orcid.org/0000-0002-7589-8471

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
