## [Reviewer comments · BMJ Open]

ARTICLE DETAILS

TITLE (PROVISIONAL)	SIZE study: study protocol of a multi-centre, randomised controlled trial to compare the effectiveness of an interarcuar decompression versus extended decompression in patients with intermittent neurogenic claudication caused by lumbar spinal stenosis.
AUTHORS	Arjun Sharma, Jamie; Gadjradj, Pravesh; Peul, Wilco; van Tulder, Maurits; Moojen, Wouter; Harhangi, Biswadjet

VERSION 1 – REVIEW

REVIEWER	Peter Försth Dept of surgical sciences Uppsala University Hospital Sweden
REVIEW RETURNED	25-Feb-2020

GENERAL COMMENTS	The authors should be honoured for planning an important study on surgical methods in the treatment of lumbar spinal stenosis. However, there are some concerns about the methodology. First, a remark about the surgery. The method called standard treatment or laminectomy as it is described in the protocol is in my view a method that has been abandoned among most spine surgeons at least a decade ago. A total resection of the two adjacent laminae for a decompression is clearly unnecessary. Many surgeons use a central approach called laminectomy with removal of the midline structures for good access to the spinal canal but with minimal resection of other stabilising structures. Some recent references on this topic are missing. E.g. the Cochrane review by Overvest et al. Effectiveness of posterior decompression techniques compared with conventional laminectomy for lumbar stenosis. 2015. Also the paper by Nerland et al in BMJ 2015. Minimally invasive decompression versus open laminectomy for central stenosis of the lumbar spine. There are some concerns about unclear inclusion criteria's and not clearly defined surgical methods. These shortages and details of remarks below are commented in the attached pdf. The follow-up of the study is very ambitious with data collection on several occasions up to 4 years post-op. 5 occasions during the first year might lead to weariness among participants and thus a risk for high loss to follow-up. I suggest a decrease in the number of follow-up. The physical examination is also very ambitious but risk not to give much information. This because findings on neurologic
--

	examination is very rare in LSS and that the two performance tests in the protocol not is designed to measure walking capacity which aside from pain is a major concern for LSS patients. I suggest the authors to consider the use of a walking test. A revision of the protocol according to these remarks is suggested before a publication in BMJ Open can be considered.
--	---

REVIEWER	Jens Ivar Brox Department of Physical Medicine and Rehabilitation, Oslo University Hospital. Medical faculty, University in Oslo, Norway.
REVIEW RETURNED	02-Mar-2020

GENERAL COMMENTS	This is a well written study protocol to address a relevant research question. The study is plannend according to guidelines and it is reported according to current standards. In my opinion the protocol have some limitations, such as: patients were not included in the planning; 12 weeks duration of symptoms is quite short to indicate surgery for spinal stenosis; the sparse evidence of the effectiveness of surgery as compared with non-operative treatment is not addressed (see Zaina et al., Cochrane 2016); reference list is biased (lack of Zaina and the protocol of Hermansen et al.2017) and the comparison does not include sham-surgery or non-operative treatment. Interestingly, a quite similar protocol was published in 2017 by Hermansen E et al. It is difficult to understand why this publication is not on the reference list. The author group must know about this study. 465 patients were included in the Norwegian trial and 2-years follow-up has been conducted. There are of course differences in the PICO's in the two trials. Most differences are of doubtful relevance and some are semantics. Both studies aim to compare the effectiveness of surgical methods for neurogenic claudication. The Norwegian study is larger and include more centres. Median age is likely to be the same in the two studies. The main inclusion criteria are symptoms and MRI in both studies. Both because of the sparse evidence of the effectiveness of surgery and that spinal stenosis on MRI is a common finding in people without symptoms, the studies should have attempted to target a more limited population. This is difficult but nevertheless a time honored question because some patients profit from surgery and some do not. Both studies compare bilateral laminotomy with another surgical method. The Dutch claim that wide bony decompression is the usual care today. In my opinion many surgeons will disagree and say that less invasive surgery is the usual care. Still the interventions compared in both trials are relevant in my opinion. The patients and observer and statistician blinded design is similar in the two protocols. In my opinion also the writing process should be blinded because of strong preferences among surgeons and possible biased interpretation of results. The primary outcome in both studies is pain related disability (RMQ and ODI). This is a PROM designed for back pain, while the
--

	indication for surgery is claudication. Non-specific changes including placebo are expected to affect ODI and RMQ more than they might influence a more specific outcome like walking distance (see Malmivara et al). Secondary outcomes do not include the more specific Zürich Claudication Index but a lot of other relevant outcomes. I would prefer a simple and face valid measure such as walking distance as compared to the more sophisticated physical tests applied. Sample size is calculated to detect a difference of 3 points which is likely to have no clinical relevance, but at the same time it means the study is adequately sized for the primary outcome but may be not for other outcomes. Planned statistical analyses are in my opinion properly reported.
--	---

REVIEWER	Irina Chis Ster St George's University of London
REVIEW RETURNED	15-May-2020

GENERAL COMMENTS	I have a general positive view on this well written manuscript. I have a question regarding the sample size calculation and the analyses plan of the primary outcome, i.e. RMDQ at 12 months after randomization. The authors state that "The results of the RMDQ will be assessed using generalized linear mixed models (GLMM), to account for the repeated measurements within patients and the multi-centre stratification." I am not entirely sure if the authors have the adequate power to employ such analyses. The sample size calculation is bluntly simple; essentially ignoring the complex structure of the data which the authors plan to account for in the analysis. I appreciate the difficulty of finding the right input parameters to feed the sample size calculation in such complex setting especially if their research is new. Perhaps they held a pilot? Perhaps they can conduct some sensitivity analyses around some numbers which the authors considered clinically sensible? Can they justify the conflict between the sample size and the analysis plan of the primary outcome?
---

VERSION 1 – AUTHOR RESPONSE

B.S. Harhangi MD, PhD, MSc

REVIEWER #1

The authors should be honoured for planning an important study on surgical methods in the treatment of lumbar spinal stenosis.

However, there are some concerns about the methodology.

1. The method called standard treatment or laminectomy as it is described in the protocol is in my view a method that has been abandoned among most spine surgeons at least a decade ago. A total resection of the two adjacent laminae for a decompression is clearly unnecessary. Many surgeons use a central approach called laminectomy with removal of the midline structures for good access to the spinal canal but with minimal resection of other stabilizing structures.

We agree with the comments of the reviewer. Still, there are quite some surgeons who are still performing a full two level laminectomy in the Netherlands. This is one of the reasons why we want to execute this study to improve evidence for the less invasive approach. But as we do understand and know that the laminectomy has been replaced with less invasive decompressions, we have replaced

the words “gold standard” with “frequently used” and “widely used” in the abstract and the introduction (lines 55 and 99).

2. Some recent references on this topic are missing. E.g. the Cochrane review by Overvest et al. Effectiveness of posterior decompression techniques compared with conventional laminectomy for lumbar stenosis. 2015. Also the paper by Nerland et al in BMJ 2015. Minimally invasive decompression versus open laminectomy for central stenosis of the lumbar spine.

Thank you for this observation. These important references have been added to the reference list.

3. The follow-up of the study is very ambitious with data collection on several occasions up to 4 years post-op. 5 occasions during the first year might lead to weariness among participants and thus a risk for high loss to follow-up. I suggest a decrease in the number of follow-up.

We agree with the reviewer that the follow up is quite ambitious. However, four years of follow-up is necessary, because the possible instability that could occur after a laminectomy may not occur within one year but more likely after three or four years after surgical decompression. The number of occasions that a patient has to be physically present in the clinic is three times (12, 24 and 48 months postoperative) more than regular care. We would like to refer to another publication of our research group which has even more follow-up moments with very low loss to follow-up¹.

4. The physical examination is also very ambitious but risk not to give much information. This because findings on neurologic examination is very rare in LSS and that the two performance tests in the protocol not is designed to measure walking capacity which aside from pain is a major concern for LSS patients. I suggest the authors to consider the use of a walking test.

We agree with the reviewer and have removed the reflexes and sensibility part of the neurologic examination. (lines 230-236) We have added the 6-Minute Walking Test to our secondary outcomes. (lines 258-264)

REVIEWER #2

This is a well written study protocol to address a relevant research question. The study is planned according to guidelines and it is reported according to current standards.

1. In my opinion the protocol have some limitations, such as: patients were not included in the planning;

We agree with this comment and will mention this in the discussion of limitations of the final report.

2. 12 weeks duration of symptoms is quite short to indicate surgery for spinal stenosis;

Thank you for this observation. However in our practice and in various studies it is customary to adhere a period of 12 weeks e.g. 3 months of neurogenic claudication as one of the inclusion criteria². For comparability to the study of Mooijen et al. we decided to choose the same period for duration of symptoms².

3. the sparse evidence of the effectiveness of surgery as compared with non-operative treatment is not addressed (see Zaina et al., Cochrane 2016);

We agree with this comment and added this trial to our references list. We have also addressed this review in the introduction. (lines 95-96)

4. reference list is biased (lack of Zaina and the protocol of Hermansen et al.2017) and the comparison does not include sham-surgery or non-operative treatment.

The trial of Hermansen et al. has been discussed in the discussion and added to our references list. (lines 441-443)

5. The primary outcome in both studies is pain related disability (RMQ and ODI). This is a PROM designed for back pain, while the indication for surgery is claudication. Non-specific changes including placebo are expected to affect ODI and RMQ more than they might influence a more specific outcome like walking distance (see Malmivara et al). Secondary outcomes do not include the more specific Zürich Claudication Index but a lot of other relevant outcomes. I would prefer a simple and face valid measure such as walking distance as compared to the more sophisticated physical tests applied. Sample size is calculated to detect a difference of 3 points which is likely to have no clinical relevance, but at the same time it means the study is adequately sized for the primary outcome but may be not for other outcomes.

Planned statistical analyses are in my opinion properly reported.

We thank the reviewer for the suggestion of adding a test for walking distance. We have added the 6-minute walking test to our secondary outcomes. (lines 258-264)

REVIEWER #3

1. I have a general positive view on this well written manuscript.

I have a question regarding the sample size calculation and the analyses plan of the primary outcome, i.e. RMDQ at 12 months after randomization. The authors state that "The results of the RMDQ will be assessed using generalized linear mixed models (GLMM), to account for the repeated measurements within patients and the multi-centre stratification."

I am not entirely sure if the authors have the adequate power to employ such analyses. The sample size calculation is bluntly simple; essentially ignoring the complex structure of the data which the authors plan to account for in the analysis.

I appreciate the difficulty of finding the right input parameters to feed the sample size calculation in such complex setting especially if their research is new. Perhaps they held a pilot? Perhaps they can conduct some sensitivity analyses around some numbers which the authors considered clinically sensible? Can they justify the conflict between the sample size and the analysis plan of the primary outcome?

This statistical analysis plan and sample size calculations were performed by our statistician K. Mauff and were recalculated and approved by the statistician of the Medical Ethical Committee of the Erasmus Medical Center Rotterdam. We are open for discussion if necessary.

PDF COMMENTS

1. "At least 12 weeks of complaints of INC, without sufficient response to conservative management"
a. A defined lowest level of disability would be appropriate in order not to include patients with minor symptoms.

We agree with the reviewer and have defined 12 weeks of complaints of INC with a minimum level of disability of Visual Analogue Scale above 30. (line 139)

2. "indication for an operation according to consensus of surgeon"

a. Unclear! Will there be a discussion among several surgeons about eligibility for each patient
There will not be a discussion among several surgeons about eligibility, but the surgeon will look at certain components: significant limitation of activities, decreased range of motion, and progressiveness of the neurogenic claudication.

3. "a partial medial facetectomy will be performed in order to maintain stability of the segments"

a. Unclear! Do the authors mean that a partial medial facetectomy maintain stability? Or that this procedure as opposed to a total facetectomy preserves stability?

Thank you for this observation. The partial medial facetectomy maintains more stability as opposed to the total facetectomy. This has been changed in the manuscript. (lines 167-168)

4. "the posterior ligaments are spared"

a. I suggest a change to "interspinous ligaments are spared."

We agree with the reviewer and have changed this in the manuscript. (line 169)

5. "a medial facetectomy will be performed in order to maintain stability of the segments"

a. As unclear as in the description of bilat decompression above. Why not partial medial facetectomy here? In my hands a bigger part of the medial facet needs to be resected in bilateral laminotomies as compared to a central approach with laminectomy to properly decompress the lateral recess
 `We agree with the reviewer that a bigger part of the medial facet needs to be resected in bilateral laminotomies as compared to a central approach with laminectomy to properly decompress the lateral recess. We have added “partial” to the facetectomy and added “to decompress neuronal structures including the exciting nerve root”. This sentence has the same explanation as comment #3. We have changed this in the manuscript. (line 177-179)

6. Neurologic examination

a. Ambitious but LSS patients show few clinical findings on neurological examination. The result is also very dependent of the examiner and varies between examiners

Thank you for this comment. We have removed the reflexes from the neurologic examination. (lines 231-235)

7. “Timed Up and Go Test (TUG) and Timed Chair-Stand Test”

a. These tests do not measure walking capacity which is a major complaint among patients with LSS. Why not use a simple walking test like 6MWT or treadmill test

We agree with this comment of the reviewer and we added the 6-minute walk test to our secondary outcomes. (lines 258-264)

8. Functional X-ray can be done in several ways and the procedure should be strictly described in a study. Standing? Sitting? Range of motion.

Functional X-rays will be performed standing AP and lateral and in maximum flexion and extension. This has been changed in the manuscript. (lines 267-268)

9. “if the operation was implemented at the wrong vertebral level”

a. How will the surgeon know that immediately after surgery?

Usually peroperative X-ray will identify a wrong level and an extra check for level and the occurrence of any listhesis will be made 6 weeks postoperatively.

10. “3 weeks, 6 weeks, 3 months, 6 months,”

a. Unnecesary many follow-ups!

Most of the PROMS will be measured online by questionnaires. There are only four follow-up moments that patients have to be actually present at the hospital. These are the appointments with the research nurses. Six weeks postoperatively is standard procedure. So eventually it is only three additional times. These follow-up moments are a lot less than other trials we have executed1.

1. Seiger A, Gadjradj PS, Harhangi BS, et al. PTED study: design of a non-inferiority, randomised controlled trial to compare the effectiveness and cost-effectiveness of percutaneous transforaminal endoscopic discectomy (PTED) versus open microdiscectomy for patients with a symptomatic lumbar disc herniation. (2044-6055 (Electronic)).

2. Moojen WA, Arts Mp Fau - Jacobs WCH, Jacobs Wc Fau - van Zwet EW, et al. Interspinous process device versus standard conventional surgical decompression for lumbar spinal stenosis: randomized controlled trial. (1756-1833 (Electronic)).

VERSION 2 – REVIEW

REVIEWER	Peter Försth Uppsala University Hospital Sweden
REVIEW RETURNED	06-Jul-2020

GENERAL COMMENTS	The authors should be honoured for planning an important study on surgical methods in the treatment of lumbar spinal stenosis. Important improvements have been made in the protocol since the first review. I persist in my remark about surgical technique. The method called “usual care: conventional laminectomy” as it is described in the protocol is in my view a method that has been abandoned among most spine surgeons. A total resection of the two adjacent laminae for a decompression is clearly unnecessary. Many surgeons use a central approach and call it laminectomy with removal of the midline structures only in the affected segment with minimal resection of other stabilising structures and no affection on the adjacent segments. The laminectomy in the protocol means a disruption of the posterior midline structures not only in the affected stenotic segment but also in the upper and lower adjacent segments. Thus 3 segments are affected. If two levels are stenotic the procedure means that 5 segments will be affected with disruption of the midline structures. Thus, the study will compare the bilateral small size interarcular decompression to a very invasive technique that in my view is not in common practice. However, the protocol is now improved with a clarification on what is actually being compared. The follow-up of the study is still a bit over ambitious with data collection from the participants on 5 occasions during the first year which means a risk of weariness among participants and thus a risk for high loss to follow-up. The addition of the 6MWT is an improvement as it measures the walking capacity which in lumbar stenosis is a main concern among patients. The protocol is now recommended for publication and the study will bring some valuable insights to the spine surgery community.
--

REVIEWER	Irina Chis Ster St George's University of London
REVIEW RETURNED	15-Jul-2020

GENERAL COMMENTS	The authors clarified that the main analysis would be conducted on their univariate primary outcome which is RMDQ at 12 months and that would be their main result. They should also clarify that the subsequent analyses of the longitudinal outcome and others are of exploratory value - they do not have sufficient the power for statistical inference using those analyses. Having said this and looking into the carefully crafted detailed plan of the analyses which include advanced concepts I do not see how this would be conducted without a professional statistician. That level of detail was not even necessary at this stage (detailed analyses plans are never necessary in the absence of the data - broad lines are sufficient) but they are all accurate and correct. Could be Katya Mauff be included in the list of authors for her valuable and competent advice - equally important as everyone else's. I hope the Editor agrees that she deserves authorship.
---

VERSION 2 – AUTHOR RESPONSE

Reviewer(s)' Comments to Author:

Reviewer: 1

Reviewer Name: Peter Försth

Institution and Country:
Uppsala University Hospital
Sweden

Please state any competing interests or state 'None declared': None declared

Please leave your comments for the authors below

The authors should be honoured for planning an important study on surgical methods in the treatment of lumbar spinal stenosis. Important improvements have been made in the protocol since the first review.

Thank you for your comments.

I persist in my remark about surgical technique. The method called "usual care: conventional laminectomy" as it is described in the protocol is in my view a method that has been abandoned among most spine surgeons. A total resection of the two adjacent laminae for a decompression is clearly unnecessary. Many surgeons use a central approach and call it laminectomy with removal of the midline structures only in the affected segment with minimal resection of other stabilising structures and no affection on the adjacent segments. The laminectomy in the protocol means a disruption of the posterior midline structures not only in the affected stenotic segment but also in the upper and lower adjacent segments. Thus 3 segments are affected. If two levels are stenotic the procedure means that 5 segments will be affected with disruption of the midline structures. Thus, the study will compare the bilateral small size interarcular decompression to a very invasive technique that in my view is not in common practice. However, the protocol is now improved with a clarification on what is actually being compared.

Thank you for your comments. We understand that the classical laminectomy has been left by many surgeons as technique. But that does not change that a laminectomy is still used in clinical practice. We made some changes to the manuscript to emphasize this.

The follow-up of the study is still a bit over ambitious with data collection from the participants on 5 occasions during the first year which means a risk of weariness among participants and thus a risk for high loss to follow-up.

We understand this concern. However in a recent completed randomized trial by our research group, focused on lumbar spine surgery, data were collected on 7 occasions and the loss to follow-up was 13% after 1 year. We expect comparable or better rates of follow-up with these data collection points.

The addition of the 6MWT is an improvement as it measures the walking capacity which in lumbar stenosis is a main concern among patients.

The protocol is now recommended for publication and the study will bring some valuable insights to the spine surgery community.

Thank you for your valuable comments.

Reviewer: 3

Reviewer Name: Irina Chis Ster

Institution and Country: St George's University of London

Please state any competing interests or state 'None declared': None declared.

Please leave your comments for the authors below

The authors clarified that the main analysis would be conducted on their univariate primary outcome which is RMDQ at 12 months and that would be their main result. They should also clarify that the subsequent analyses of the longitudinal outcome and others are of exploratory value - they do not have sufficient the power for statistical inference using those analyses.

We made the corrections as suggested.

Having said this and looking into the carefully crafted detailed plan of the analyses which include advanced concepts I do not see how this would be conducted without a professional statistician. That level of detail was not even necessary at this stage (detailed analyses plans are never necessary in the absence of the data - broad lines are sufficient) but they are all accurate and correct. Could be Katya Mauff be included in the list of authors for her valuable and competent advice - equally important as everyone else's. I hope the Editor agrees that she deserves authorship.

To qualify for authorship, we adhere to the ICMJE-criteria. We clarified this at both the end and beginning of the manuscript. Katya Mauff gave advice and only giving advice merits to be acknowledged as contributor and not as co-author. Ms. Mauff has also received financial compensation for her advice.